# A problem of Epic proportion

Rawan Abulibdeh[1], Matthew G. Crowson[2,3]☯, Molly J. Douglas[4]☯, Mena Ramos[5]☯, Noelle N. Saillant[6,7], Leo Anthony Celi [8,9,10]¤*

1 Department of Electrical and Computer Engineering, University of Toronto, Toronto, Ontario, Canada, 2 Department of Otolaryngology–Head & Neck Surgery, Harvard Medical School, Boston, Massachusetts, United States of America, 3 Department of Otolaryngology–Head & Neck Surgery, Mass General Brigham, Boston, Massachusetts, United States of America, 4 University of Arizona, Tucson, Arizona, United States of America, 5 Global Ultrasound Institute, University of California San Francisco, San Francisco, California, United States of America, 6 Department of General Surgery, Boston University Chobanian & Avedisian School of Medicine, Boston, Massachusetts, United States of America, 7 Division of Trauma, Acute Care Surgery & Surgical Critical Care, Boston Medical Center, Boston, Massachusetts, United States of America, 8 Laboratory for Computational Physiology, Massachusetts Institute of Technology, Cambridge, Massachusetts, United States of America, 9 Division of Pulmonary, Critical Care and Sleep Medicine, Beth Israel Deaconess Medical Center, Boston, Massachusetts, United States of America, 10 Department of Biostatistics, Harvard T.H. Chan School of Public Health, Boston, Massachusetts, United States of America

¤ Laboratory for Computational Physiology, Massachusetts Institute of Technology, Cambridge, Massachusetts, United States of America
☯ These authors contributed equally and are listed alphabetically by last name.
* lceli@mit.edu

## Abstract

In the United States today, one private company holds the digital keys to the nation's health. Epic Systems provides the electronic health record for 42.3% of acute care hospitals and controls over half (54.9%) of all acute care hospital beds, a concentration of market power unprecedented in modern healthcare IT. This paper investigates how Epic transformed from a small, privately held vendor into the dominant force shaping the electronic health record landscape, drawing on peer-reviewed literature, federal antitrust filings, and cross-national case studies. We chart its ascent from a unified-platform niche player, to market validation through Kaiser Permanente's multi-billion-dollar rollout, to rapid adoption under the HITECH Act. Quantitative analysis shows the U.S. hospital electronic health record market shifting from competitive to highly concentrated (Herfindahl–Hirschman Index > 2,500) after 2018, with Epic capturing nearly 70% of new hospital contracts in 2024. We examine the reinforcing mechanisms and standard practices, including network effects that add true interoperability value, high switching costs common to enterprise software, bundling practices, and workforce restrictions, that entrench this dominance, alongside alleged anti-competitive behaviors under active litigation. While some consolidation reflects legitimate business efficiencies and the inherent complexity of healthcare IT, the current level of market concentration poses significant governance challenges. International deployment failures in Norway, Denmark, Finland, and the United Kingdom reveal that Epic's success reflects market structures rather than clear technological

**Data availability statement:** All data underlying the findings in this study are derived from publicly available sources and are fully cited within the manuscript. No new datasets were generated or analyzed for this study, and no proprietary or restricted data were used. The information provided in the manuscript is sufficient to replicate all reported findings.

**Funding:** The authors received no specific funding for this work. LAC receives research funding from the National Institute of Health through DS-I Africa U54 TW012043-01 and Bridge2AI OT2OD032701, and the National Science Foundation through ITEST #2148451, a grant of the Boston-Korea Innovative Research Project (RS-2024-00403047) and a grant of the Korea Health Technology R&D Project (RS-2024-00439677) through the Korea Health Industry Development Institute (KHIDI) as funded by the Ministry of Health & Welfare, Republic of Korea. These funding sources did not support the work described in this manuscript. The funders had no role in study design, data collection and analysis, decision to publish, or preparation of the manuscript.

**Competing interests:** I have read the journal's policy and the authors of this manuscript have the following competing interests: Dr. Crowson reports serving as Chief Medical Officer of Drive Health, a digital health company. Dr. Saillant reports consulting relationships with Zimmer Biomet and Haemonetics. All other authors declare no competing interests.

superiority. Regulatory contrasts highlight how the European Health Data Space's mandatory interoperability, portability, and pre-market testing requirements are designed to prevent similar monopolization in the EU. We conclude with a reform blueprint that spans antitrust enforcement, structural separation, public utility models, and clinician-driven redesign, arguing that electronic health record systems must be governed as essential public health infrastructure. Confronting Epic's monopoly is critical to restoring competition, fostering innovation, and ensuring that digital health serves patients and the public good.

## Author summary

Electronic health records (EHRs) are the digital backbone of modern healthcare. They store patient information, support clinical decisions, and enable data sharing across health systems. In the United States, however, this essential infrastructure is now dominated by a single private vendor, raising important questions about competition, interoperability, and public accountability. In this study, we examine how Epic Systems came to control a majority of U.S. hospital beds and why this level of market concentration matters. Using publicly available market data, policy analysis, legal filings, and international case studies, we show that Epic's dominance is not simply the result of superior technology. Instead, it reflects structural forces such as federal incentive programs, weak interoperability requirements, high switching costs, and network effects that reinforce vendor lock-in. We compare the U.S. experience with large-scale EHR deployments in Europe, where different regulatory approaches have shaped market outcomes. We conclude by outlining policy options to reduce the risks of concentrating essential health data infrastructure in private hands and argue that EHR governance is a matter of public interest with direct consequences for patients, clinicians, and the future of digital health.

## Introduction

Epic Systems has established what is arguably the most entrenched monopoly in the history of healthcare information technology. As of 2024, its platform serves 42.3% of U.S. acute care hospitals and 54.9% of hospital beds [1], granting it unparalleled control over the digital backbone of American healthcare. By 2018, the U.S. electronic health record (EHR) market had already surpassed the Federal Trade Commission's threshold for "highly concentrated" industries, with a Herfindahl–Hirschman Index above 2,500 [2]. This consolidation has fueled a series of exclusionary practices now under scrutiny in landmark federal antitrust cases, including *CureIS v. Epic* and *Particle Health v. Epic*, which allege an "Epic-first" policy requiring customers to adopt Epic products where available and practices that make it "commercially impossible" for rivals to access patient records stored in Epic's systems [3,4].

Yet Epic's position rests less on technical superiority than on the convergence of early strategic choices, structural market dynamics, and permissive regulation—conditions that together entrenched high switching costs, proprietary data architectures, and self-reinforcing network effects [5,6]. These barriers have not only insulated the company from competition but also extended its reach beyond vendor choice, enabling it to dictate how medical data is captured, exchanged, and monetized, and in turn to shape the infrastructure on which care delivery, biomedical research, and health policy increasingly depend [5,6]. While defenders cite Epic's scalability, successful large-scale implementations, and seamless interoperability within its own ecosystem [7,8], independent evaluations tell a more complex story. Studies of major deployments have consistently reported workflow challenges, usability concerns, and persistent clinician dissatisfaction [9–11]. Epic's commercial dominance, thus, reflects not only product capability but also structural forces that reinforce its position. When one company controls the digital backbone of care delivery, it not only influences market outcomes but also sets the terms of public health access itself [12].

At its core, an electronic health record is the operational spine of modern healthcare, responsible for documenting clinical encounters, storing and transmitting patient data, orchestrating orders and results, and coordinating the workflow of every clinician who touches a patient. The central concern is not merely that a single vendor dominates the electronic health record market, but that critical health infrastructure has become privatized with minimal public oversight. When a single company controls the flow of patient information, foundational decisions about access, interoperability, and innovation are guided by private incentives rather than public needs. This privatization of the digital backbone of healthcare raises profound questions of accountability, equity, and long-term resilience; issues that demand policy attention far beyond traditional market regulation.

This paper examines the structural, economic, and regulatory forces that enabled Epic's rise, from federal incentives like the HITECH Act that accelerated adoption without guarding against market concentration, to hospital consolidation and weak interoperability mandates that entrenched proprietary control over what now functions as a public utility. Drawing on legal filings, market share data, and comparative international models, it situates Epic's dominance as not merely a market 'success' but a governance challenge, one that requires targeted policy reform, from antitrust evaluation and interoperability mandates to the development of public-interest guardrails around essential health data infrastructure. Without such measures, the U.S. risks locking in private control over critical health infrastructure for decades to come, with lasting consequences for innovation, competition, and patient autonomy.

## The rise of an empire

Epic's path to market dominance began with an unorthodox foundation. Founded in the 1970s in Madison, Wisconsin, Judy Faulkner rejected venture capital and resisted going public, insulating the company from short-term shareholder pressures and enabling decades-long strategic horizons [13,14]. From the outset, Epic pursued a unified-platform architecture, delivering tightly integrated modules in an industry where competitors expanded through fragmented acquisitions. This design choice not only streamlined interoperability within Epic's ecosystem but also created high switching costs that became structural advantages as its footprint expanded. [5].

Epic's breakthrough came in 2003 with Kaiser Permanente's selection of its system for a $1.8 billion deployment for its 8.4 million patients [15–17]. This contract demonstrated Epic's capacity to manage large-scale integrated delivery networks and established it as a viable competitor in the enterprise EHR market, marking a significant milestone for the company. The 2009 Health Information Technology for Economic and Clinical Health (HITECH) Act provided the most significant acceleration in Epic's history. By allocating $27 billion in EHR adoption incentives, and introducing penalties for non-adopters beginning in 2015 [18–20], HITECH transformed certified EHR adoption among non-federal acute care hospitals increased from roughly 10% in 2008 to 96% by 2021 [21]. Epic's comprehensive platform, proven implementation record, and established large-system footprint positioned it to capture outsized market share from this adoption wave, meanwhile weak certification standards that failed to test real-world performance allowed inferior systems to enter the market, pushing hospitals to consolidate around better-performing vendors such as Epic [2,22].

Epic's customer acquisition strategy reinforced these advantages. Rather than relying on traditional sales and marketing approaches, Epic built its market position through word-of-mouth referrals and customer satisfaction [23,24], underpinned by a reputation for reliable implementations. In an industry facing persistent implementation challenges and adoption barriers [25], Epic's reputation for reliable implementations, particularly in the US market, became a powerful signal of risk reduction. For large health systems facing the immense financial, operational, and reputational stakes of an EHR transition, Epic's track record of rigorous implementation methodology created a self-reinforcing moat around its customer base [24].

## From growth to market capture

The consolidation of the U.S. EHR market reflects multiple forces, many not deliberately anti-competitive, yet the resulting concentration has created conditions that now pose clear antitrust and governance concerns. Healthcare IT faces inherent challenges that favor scale: the extraordinary complexity of clinical workflows, the need for continuous regulatory compliance updates, the high fixed costs of development and maintenance, and technical advantages of unified platforms for care coordination. Large health systems may prefer vendors capable of managing enterprise-wide implementations across diverse care settings, and Epic's proven track record in complex deployments addressed real institutional needs [5,24]. The 2009 HITECH Act created urgent adoption timelines that rewarded vendors with existing capacity and implementation experience. Hospital consolidation driven by payment reform, economies of scale, and competitive pressures pushed toward EHR standardization within merged entities. Network effects in health information exchange, while reinforcing Epic's dominance, also delivers genuine value through improved care coordination [6,26]. The policy challenge is not that consolidation occurred, but that it proceeded without adequate safeguards against monopolization of what has become essential public infrastructure. Understanding these legitimate drivers are crucial for designing reforms that preserve efficiencies while addressing anticompetitive harms.

By the early 2010s, the U.S. EHR market was still relatively fragmented, with multiple vendors, including Cerner, CPSI, Meditech, McKesson, and others, competing across hospitals of varying sizes and complexities [2]. Even the largest players held only modest market shares, and purchasing decisions were often driven by cost, usability, and existing relationships. That equilibrium shifted rapidly. Between 2012 and 2018, the Herfindahl–Hirschman Index for the national EHR market rose from 1,452 ("competitive") to over 2,500 ("highly concentrated") [2]. Epic's share of U.S. acute care beds more than doubled in that period from 20.6% to 46.5% while Epic and Cerner together came to control over 70% of all beds nationally.

Consolidation has continued to accelerate, with Epic recording its largest net gain in 2024, adding 176 hospitals and 29,399 beds, and capturing nearly 70% of all hospital EHR decisions across institutions of every size [1]. Epic Systems now manages electronic health records for over 305 million patients worldwide, making it the most widely adopted EHR platform and a dominant fixture within the U.S. healthcare system [1,27]. Of the 20 hospitals on U.S. News & World Report's 2025–26 Best Hospitals Honor Roll, nearly all are known to use Epic Systems for their electronic health records, reflecting the vendor's dominant market share among the nation's top-ranked hospitals [27–29].

Epic's technical and structural properties strategically reinforce this trajectory through a deeply integrated, proprietary architecture that creates substantial switching costs, amplified by network effects that favor market leaders [30,31]. Hospitals are more than five times as likely to exchange patient data when they share the same vendor, demonstrating quantified network effects that create competitive advantages for dominant platforms [6]. The company's "Care Everywhere" platform processes over 20 million patient records daily, creating interoperability advantages that increase with market share [6,26]. As Epic's installed base grows, so too does the value, and the lock-in, of remaining within its ecosystem.

Epic's dominance in academic centers magnifies these effects. Roughly 60% of U.S. teaching hospitals now run Epic [29,32], and more than 90% of medical students and residents train on its systems [27]. This early exposure creates a generational lock-in where new clinicians enter the workforce already fluent in Epic's workflows, lowering adoption costs

for hospitals and diminishing the appeal of alternatives. More recently, Epic has expanded into rural markets, historically the stronghold of lower-cost, modular systems, further widening the competitive gap [1].

What began as a multi-vendor marketplace has, in little more than a decade, consolidated into a structure where one company's technical roadmap and commercial priorities can influence national data governance, interoperability standards, and the pace and direction of health IT innovation.

## The case for Epic's dominance

Before examining the mechanisms that maintain Epic's market position, we need to consider arguments in its defense, from customers and industry analysts, to the company itself. These perspectives hold that Epic's dominance reflects merit rather than monopolization, and that market concentration delivers benefits that critics overlook. Epic's defenders emphasize its superior implementation methodology and customer support model. Unlike competitors that rely heavily on third-party consultants, Epic maintains tight control over implementations through its own staff, which proponents argue ensures consistency and accountability [23,24]. The company's Intensive training programs and ongoing customer support have generated high satisfaction scores, with KLAS Research consistently ranking Epic among the top vendors for customer experience [1]. Hospitals report that Epic's unified platform eliminates integration headaches common with multi-vendor environments, and that its comprehensive feature set reduces the need for third-party add-ons [7,33]. The network effects that critics view as anticompetitive are, in this view, a natural and beneficial outcome of widespread adoption. Epic's Care Everywhere network facilitates over 20 million patient record exchanges daily [26], enabling seamless care coordination across health systems—a genuine clinical benefit that grows with scale. Defenders argue that interoperability works best when providers share common platforms, and that Epic's market share reflects hospitals' recognition of this reality. The company's investments in Fast Healthcare Interoperability Resources (FHIR) APIs and participation in frameworks like the Trusted Exchange Framework and Common Agreement (TEFCA) [34] demonstrate commitment to open standards, even if critics contend these efforts are insufficient. From a business perspective, Epic's pricing and bundling practices can be defended as standard for enterprise software. Integrated platforms are more expensive but deliver value through reduced complexity, single-vendor accountability, and economies of scope. The company's profitability [35,36] reflects efficient operations and valuable products, not monopoly rents, according to this view. Epic's resistance to private equity and public markets has allowed it to take longer-term views on product development rather than maximizing short-term returns [13,14]. Perhaps most importantly, defenders argue that Epic's success reflects healthcare organizations' voluntary choices in a competitive market. No law requires hospitals to choose Epic; they do so after extensive evaluation processes that consider alternatives. The fact that prestigious academic medical centers, sophisticated health systems, and even competitors' former customers repeatedly select Epic suggests the product delivers value commensurate with its cost [27–29]. High switching costs exist for any complex enterprise system, not uniquely for Epic. If hospitals truly found Epic's product inadequate or its practices predatory, they would migrate to competitors despite the costs. These arguments have merit and explain why Epic maintains strong customer loyalty. The question is not whether Epic provides value to its customers—evidence suggests it often does—but whether its market dominance creates systemic problems that extend beyond individual customer satisfaction. A vendor can deliver quality products while simultaneously exercising market power in ways that harm competition, innovation, and the broader healthcare ecosystem. The challenge for policymakers is distinguishing legitimate competitive success from anticompetitive behavior, and determining when market concentration itself, regardless of any individual company's conduct, requires intervention to protect the public interest.

## Mechanics of market control

Epic's market dominance reflects multiple reinforcing mechanisms, some resulting from strategic business decisions and others from structural characteristics common to enterprise software markets. Steep economic barriers, proprietary technical architectures, powerful network effects, bundling practices, and workforce restrictions collectively generate a

self-reinforcing cycle, where each new adoption further entrenches the structural barriers to exit. While some of these mechanisms are standard business practices, their cumulative effect in a highly concentrated market raises significant policy concerns.

### Economic barriers

One of the most formidable deterrents to switching away from Epic is cost. For large health systems, migration expenses range from hundreds of millions to over $1 billion, a figure that encompasses licensing, consulting, workflow redesign, and data conversion [5,37,38]. Beyond financial costs, hospitals also face significant productivity losses during transitions as staff adapt to new workflows [39]. These challenges are exacerbated by the scarcity of Epic-certified consultants and trainers, creating dependence on a limited pool of specialists to execute migrations that drive up costs and logistical complexity. This creates a double bind since even organizations prepared to absorb the financial hit often lack the workforce capacity to execute a successful switch.

### Technical lock-in

Epic's architecture has been deliberately engineered to create dependence. The system is proprietary end-to-end, requiring Epic-certified staff for configuration and integration, which prevents hospitals from cultivating independent internal expertise [40,41]. While Epic outwardly complies with FHIR standards [42], its implementations maintain dependencies that preserve competitive advantages. Data flows seamlessly between Epic hospitals but become fragmented and less integrated when exchanged with non-Epic vendors [43,44]. At scale, this imbalance means that interoperability works best for those already in the Epic ecosystem, which creates strong disincentives for hospitals to exit.

### Network effects

Epic's Care Everywhere network now processes over 20 million patient record exchanges daily [26], with research indicating interoperability scores of 0.68 when healthcare sites share the same vendor compared to 0.22 across different vendor systems. [45]. This disparity generates a powerful network effect where the more hospitals that adopt Epic, the greater the incentive becomes for others to join, since remaining outside means enduring inferior interoperability. Clinicians trained in Epic environments also contribute to this effect. As Epic dominates teaching hospitals [29,32], new physicians enter practice already fluent in its workflows [27], making Epic facilities more attractive workplaces and reinforcing its centrality in clinical operations.

### Bundling & pricing strategies

Epic's sales model centers on an integrated platform, with healthcare organizations commonly adopting multiple modules such as patient portals, billing systems, and analytics tools to achieve full operational integration [33,46]. While this offers customers convenience and operational efficiency, it also creates deeper dependencies on Epic's ecosystem, potentially increasing switching costs and complexity when considering third-party alternatives. Notably, Epic has maintained premium pricing, reflected in revenue growth from $3.3 billion in 2020 to $4.9 billion in 2023 [35] and estimated earnings before interest, taxes, depreciation, and amortization (EBITDA) margins exceeding 30% [36], despite persistent criticisms about usability. These costs are tolerated not primarily for superior usability, but largely due to Epic's market entrenchment and integration advantages, which make alternatives increasingly impractical for most health systems.

### Workforce restrictions

Finally, Epic has historically utilized non-compete agreements affecting almost 4,500 Epic-linked firms [47]. These contractual provisions, where enforceable, may limit experienced Epic personnel from joining rivals or assisting health systems

seeking to migrate away from Epic's platform. It should be noted that the enforceability of non-compete agreements varies significantly by state, and such agreements have faced increasing legal challenges and regulatory scrutiny. The Federal Trade Commission proposed a rule in 2023 to ban most non-compete clauses, though its implementation remains subject to legal challenge [48]. Nevertheless, even where such agreements face enforceability questions, they may create practical barriers that constrain labor mobility. By concentrating specialized expertise within its ecosystem, Epic maintains structural advantages that reinforce its market position.

## Allegations of anti-competitive behavior

Two major federal antitrust lawsuits filed in 2024–2025 contain allegations that, if proven, would demonstrate significant anti-competitive conduct. In September 2024, *Particle Health v. Epic Systems Corporation* [4], was filed in the Southern District of New York, alleging that Epic made it "commercially impossible" for competitors to access patient records within its systems through a series of exclusionary tactics designed to suppress competition in adjacent payer platform markets. According to the complaint, Epic cut off data access to Particle's customers for more than six months, disrupting patient care and eroding trust in Particle's services. The lawsuit further alleges that Epic submitted what Particle characterizes as 'now-discredited' security complaints against Particle within the Carequality network and inundated its support systems with what Particle describes as baseless security concerns, diverting resources and impairing client service. Most critically, Particle's complaint asserts that Epic blocked access to over 2,800 oncology patient records, allegedly leaving clinicians without timely information and directly compromising patient care. Epic has denied these allegations and the case remains pending. The ultimate determination of these claims will be made by the court.

In May 2025, *CureIS Healthcare v. Epic Systems Corporation* was filed in the Northern District of California, alleging unlawful product tying and interference with customer contracts [3,49,50]. The complaint centers on what CureIS characterizes as an 'Epic-first policy,' which it alleges requires customers already using Epic EHR or revenue cycle products to adopt Epic solutions in other categories whenever available. CureIS contends that Epic coerced hospitals into replacing CureIS modules with Epic alternatives regardless of performance, and pressured clients to terminate long-standing CureIS contracts, resulting in the alleged loss of at least seven customers. The lawsuit further alleges that Epic obtained proprietary product specifications under false pretenses and promoted non-existent forthcoming solutions as a tactic to dissuade hospitals from contracting with competitors. Epic has disputed these characterizations and the litigation is ongoing. These allegations, if substantiated, would represent significant violations of antitrust law.

*Important note on pending litigation:* The allegations detailed in *Particle Health v. Epic* and *CureIS v. Epic* represent claims made by the plaintiffs and have not been adjudicated. Epic has denied wrongdoing in both cases, and the legal proceedings are ongoing. This manuscript presents these allegations as evidence of concerns raised about Epic's market conduct, not as established facts. The ultimate determination of whether these practices violate antitrust law or other legal standards will be made through the judicial process. Regardless of litigation outcomes, however, the underlying market structure—characterized by high concentration, significant switching costs, and powerful network effects—warrants policy attention independent of any individual company's conduct.

## Failed competitive attempts (domestic & global)

Once Epic's chief rival, Cerner (acquired by Oracle in 2022) has steadily lost ground, shedding 74 hospitals and approximately 17,000 hospital beds to Epic in 2024 alone as its share of the acute care market continued to erode [1]. Allscripts, a decade ago a leading hospital EHR vendor, succumbed to similar pressures, selling off its hospital and large physician practice business in 2022 and effectively exiting the acute care market [51]. Even where hospitals have attempted to leave Epic, transitions have proven prohibitively expensive. Partners HealthCare (now Mass General Brigham), for example, initially budgeted $600 million for its Epic rollout but ultimately spent $1.2 billion once consultant fees, training, lost revenues, and expanded staffing were accounted for [52,53].

While Epic dominates the U.S. market, its international track record shows that its success is far from universally replicable. Across Europe, several high-profile deployments have encountered significant challenges, but attributing these difficulties solely to Epic's product would be an oversimplification. Large-scale EHR implementations are notoriously difficult regardless of vendor, involving massive organizational change, workflow redesign, and cultural adaptation [10,25]. However, the patterns observed in Norway, Denmark, Finland, and the United Kingdom suggest specific mismatches between Epic's U.S.-oriented workflows and governance assumptions and the organizational, regulatory, and clinical environments of European health systems. Rather than demonstrating Epic's technical superiority or the inevitability of its model, these cases highlight how dependent its performance is on the structural conditions that characterize the U.S. market.

In Norway, the NOK 3.7 billion Helseplattformen project produced widespread clinician protests, and reports from health authorities warning that the system endangered patient safety and exceeded projected costs [9,10,54]. However, the project involved not just installing Epic's software but fundamentally reorganizing clinical workflows across multiple hospitals with different existing systems and practices. Translation and localization challenges were significant, as Epic's platform was originally designed for U.S. healthcare contexts. Similar patterns emerged. In Denmark, the DKK 2.8 billion Sundhedsplatformen rollout [55] sparked confusion and usability failures, such as mistranslated prompts that prolonged the time required to complete routine clinical tasks [10]. Physician satisfaction remains low, with nearly one-third reporting deep dissatisfaction even years after launch [10]. Finland's €408 million Apotti system faced similar challenges [56,57]. Fewer than 10% of clinicians agreed that it improved care quality, and academic evaluations found organizations remained locked in a reactive mode of crisis management rather than realizing promised benefits [10]. At Cambridge University Hospitals in the United Kingdom, a £200 million Epic implementation led to emergency designations, ambulance diversions, and senior executive resignations [11,58].

## Structural amplifiers beyond Epic

Epic's entrenchment is not solely the product of its own strategies. Broader structural dynamics in U.S. healthcare have magnified its reach and reinforced its quasi-monopolistic control, making adoption not merely advantageous but, in many cases, functionally unavoidable. When hospital systems merge, executives increasingly consolidate on a single enterprise-wide EHR to streamline governance and minimize interoperability friction. By 2016, nearly 90% of U.S. metropolitan areas already had highly concentrated hospital markets, creating procurement environments predisposed to favor large, entrenched vendors [59]. This consolidation extends beyond hospitals into physician practice acquisitions. As independent practices are absorbed into health systems, EHR decisions shift from small-group autonomy to enterprise-level procurement [60,61]. This has the effect of pulling thousands of physicians into Epic's orbit not through head-to-head vendor competition, but through ownership realignments that reallocate IT decision-making power.

Payment structures in U.S. healthcare further reinforce Epic's dominance. Because reimbursement depends heavily on meticulous documentation, health systems prioritize EHRs optimized for billing and regulatory compliance [62]. Epic has built a strong reputation in revenue cycle management and coding efficiency, aligning its platform with financial imperatives [40,63]. These dynamics create a reinforcing feedback loop where hospital consolidation pushes systems to standardize, vertical integration extends Epic into outpatient care, and payment incentives reward the documentation-heavy workflows that have become its hallmark.

## Regulatory gaps

In the U.S., the HITECH Act of 2009 was pivotal, driving hospital adoption rates from roughly 10% in 2008 to more than 85% by 2015 [18,19]. Yet its design prioritized speed over market structure. Incentives rewarded certified EHR adoption, but early certification standards set the bar too low, allowing vendors ill-prepared for later-stage complexity to enter the market and inadvertently paving the way for consolidation around better-resourced firms. In response to these unintended consequences, the 21st Century Cures Act (2016) [64] introduced information-blocking provisions intended to promote

interoperability [65–67]. Although vendors face penalties of up to $1 million per violation [68,69], public transparency around enforcement actions remains minimal. Epic, in particular, has complied with the letter of the law while preserving its competitive advantages, for instance, by implementing FHIR standards in ways that still tether hospitals to its proprietary ecosystem [42].

The next major federal effort was TEFCA, which sought to standardize nationwide health information exchange [70]. Yet, participation is voluntary, and compliance requires little more than basic data exchange rather than true interoperability [71]. Epic remains technically compliant while ensuring that data flows most seamlessly within its own Care Everywhere network, reinforcing the lock-in dynamics already at play. Despite clear evidence of rising concentration, no major antitrust enforcement actions have targeted Epic to date. This regulatory inaction has effectively signaled tolerance for consolidation, allowing Epic's dominance to harden into structural dependency.

In stark contrast, the European Union's European Health Data Space (EHDS) regulation, adopted in 2025, was designed explicitly to prevent single-vendor lock-in [72]. It mandates standardized interoperability modules, rigorous pre-market testing of EHR systems, and continuous compliance with harmonized specifications [73,74]. Vendors are barred from marketing systems in Europe unless they first demonstrate interoperability in regulated test environments. By embedding these safeguards, the EU aims to directly block the kind of proprietary enclosures that underpin Epic's dominance in the United States. However, important caveats apply. The EHDS regulation only entered into force in 2025, and its effectiveness remains untested at scale. Implementation timeline extends through 2027–2030 [73] and the practical challenges of harmonizing health IT infrastructure across 27 member states with diverse healthcare systems, languages and vendors should not be underestimated. The ambitious pre-market testing requirements may create their own barriers to innovation and market entry, particularly for smaller vendors. Early evidence of compliance costs, vendor adaptation challenges, or unintended consequences is not yet available. EHDS represents a fundamentally different regulatory philosophy that prioritizes interoperability and data portability as legal requirements rather than voluntary market outcomes.

These differences reflect a deeper philosophical divide. U.S. policy has largely treated health data as a commercial asset managed by private vendors, whereas European regulation frames it as a public good requiring proactive governance. Whether the EHDS successfully prevents Epic-style consolidation across Europe's fragmented national healthcare systems will become clearer as implementation proceeds over the coming decade. For now, it offers a regulatory model worth monitoring as the U.S. considers its own policy responses to EHR market concentration.

## Paths to reform

Confronting Epic's dominance requires more than incremental policy adjustments. The failures of past regulation illustrate that adoption incentives and voluntary interoperability frameworks are insufficient to counter systemic lock-in. What is needed is a layered reform agenda that combines short-term regulatory action with medium-term structural redesign and long-term reconceptualization of electronic health records as critical public infrastructure (Table 1).

## Challenges and trade-offs in reform

The reform agenda outlined below must be understood in context. Healthcare IT consolidation reflects the tension between the desire for genuine efficiencies alongside competitive concerns, thus reforms must preserve legitimate benefits while addressing harms. Moreover, any significant intervention in this market carries risks of unintended consequences: disrupting existing clinical workflows, imposing transition costs on already-strained healthcare organizations, creating new barriers to innovation, or simply shifting market power rather than distributing it more equitably. The proposals below vary widely in their implementation difficulty, capital requirements, political feasibility, and time horizons. Some could be implemented relatively quickly through existing regulatory authority, while others would require new legislation, substantial funding, or multi-year coordination across federal and state governments. We present them as a menu of options requiring further analysis, stakeholder input, and pilot testing rather than as a fully specified roadmap. The goal

**Table 1. Summary of proposed reform approaches.**

| Time horizon | Primary lever | Illustrative measures | Policy objective |
|---|---|---|---|
| **Immediate (1–2 years)** | Enforcement of existing law | Antitrust scrutiny of tying and information blocking; public and punitive Cures Act penalties; mandatory participation in national data-exchange frameworks for dominant vendors; review of hold-harmless clauses and restrictive contracts. | Curb exclusionary conduct and unblock data flows without disrupting current operations. |
| **Medium-term (3–5 years)** | Market structure and incentives | Structural separation of core EHR from hosting and analytics; federal investment in shared data-exchange infrastructure; tying certification and reimbursement to usability, data quality, and clinician experience; increased patient-facing documentation and data-correction workflows. | Reduce lock-in and realign incentives toward clinical value and interoperability. |
| **Long-term (5–10 years)** | Governance of EHRs as infrastructure | Utility-style oversight for dominant vendors; federal support for open-source modular components; funding for usability-focused next-generation systems; broader governance models that incorporate clinicians, patients, and human-factors expertise. | Treat EHRs as essential public infrastructure and promote a more competitive, innovation-driven ecosystem. |

is to catalyze a serious policy debate about treating EHR systems as essential public infrastructure requiring appropriate governance, not to provide definitive solutions to problems that have challenged policymakers for decades.

## Immediate interventions (1–2 Years)

The most urgent priority is to evaluate existing antitrust frameworks as relevant to Epic's exclusionary practices. Evidence presented in *Particle Health v. Epic* and *CureIS v. Epic* provides actionable grounds for regulatory intervention under the Sherman and Clayton Acts. The Federal Trade Commission and Department of Justice should pursue enforcement against information blocking, tying arrangements, and contractual interference, treating Epic's conduct with the same seriousness historically applied to monopolies in telecommunications and technology. In parallel, the Office of the National Coordinator should strengthen enforcement of the 21st Century Cures Act's information-blocking rules by making penalties both transparent and punitive. Fines of up to $1 million per violation remain largely theoretical without publicized enforcement actions that deter vendors from undermining interoperability. Moreover, while Epic has joined the Trusted Exchange Framework and Common Agreement through its Epic Nexus network [34], participation remains voluntary. Requiring mandatory participation for vendors above a defined market-share threshold would prevent dominant players from exploiting voluntary compliance or minimal baseline exchange standards to preserve proprietary advantages. Without such a mandate, Epic can remain formally compliant while continuing to optimize data flows within its Care Everywhere network, further reinforcing the lock-in dynamics already at play. Alternatively, regulation to require open-source sharing of proprietary data models, and to govern utilization of existing standards such as HL7 FHIR, could drive true vendor-agnostic interoperability. Comprehensive underlying data standards and semantic interoperability would have the potential to fast-track new entrants into the market to reduce both lock-in dynamics and the data access disruption associated with switching systems. Finally, regulators should also explore product liability and contractual reform. Epic's hold harmless clauses, which shield it from responsibility for patient harm caused by software defects or design flaws, should be subject to legal challenge. When preventable deaths or serious injuries stem from EHR usability or access failures, liability should extend to vendors as well as clinicians and hospitals.

## Medium-term reforms (3–5 Years)

Epic's vertical integration enables it to extend EHR dominance into adjacent markets such as hosting, analytics, and revenue cycle management. Policymakers should consider structural separation, requiring vendors to decouple core EHR platforms from data hosting and ancillary services. This approach would mirror historic utility regulation in telecommunications and energy, ensuring no single firm controls the entire digital pipeline. However, structural separation poses

implementation challenges that require careful consideration. Defining the boundaries between "core EHR" and "ancillary services" is technically complex, as many functions are deeply integrated for legitimate clinical and operational reasons. Forced separation could disrupt existing customer workflows, create new interoperability challenges between separated components, and impose substantial transition costs on healthcare organizations already constrained by thin margins. Historical precedents from other industries show that separation can take years to implement, requires sophisticated ongoing regulatory oversight, and may produce unintended market distortions [75,76]. The effectiveness of separation also depends on maintaining competitive markets in the separated segments. If a single vendor dominates both EHR and ancillary markets after separation, the intervention achieves little. Despite these challenges, structural separation merits serious exploration, particularly for the largest vendors where vertical integration most clearly impedes competition. Pilot programs or gradual implementation tied to market share thresholds could allow policymakers to assess effectiveness while managing transition risks.

At the same time, federal investment in shared, public infrastructure for health data exchange (analogous to utilities or broadband backbones) could provide baseline interoperability services and reduce dependence on private vendor networks like Care Everywhere. This approach faces its own challenges, including substantial capital requirements, the need for ongoing federal maintenance and updates, and the difficulty of achieving stakeholder consensus on technical standards and governance structures. Payment reform is also critical. As long as reimbursement models reward exhaustive documentation, vendors will continue to optimize for billing rather than usability. Linking certification and reimbursement to measures of usability, data quality, and clinician satisfaction would help realign incentives toward patient outcomes and reduce the structural forces reinforcing Epic's dominance. Equally important is reforming documentation practices and strengthening patient involvement. Clinicians should reexamine what information meaningfully supports patient outcomes, rather than documenting primarily for billing compliance. Patient-facing clinical notes, where patients review and correct records in real time, may reduce errors and prevent harms caused by mis-documentation.

## Long-term transformation (5–10 Years)

Given their role as the backbone of healthcare delivery, EHR systems should be governed more like public utilities than proprietary software products. This would involve subjecting dominant vendors to oversight boards, rate-setting mechanisms, and universal service obligations to ensure equitable access.

Public utility regulation of EHR vendors would represent a fundamental departure from current practice and may face implementation hurdles. Unlike traditional utilities with relatively stable technology and clear service definitions, health IT evolves rapidly, requiring regulatory frameworks flexible enough to accommodate innovation while preventing abuse. Rate-setting for complex, customized software implementations is far more challenging than regulating standardized electricity or water services. Utility regulation typically requires substantial state capacity such as expert regulatory staff, ongoing monitoring systems, and enforcement mechanisms, and that may not currently exist in health IT oversight bodies. International experience with EHR regulation is limited, providing few tested models to draw upon. The risk of regulatory capture, where dominant vendors shape rules to their advantage, is particularly acute in technically complex industries. There is also tension between utility-style regulation (which often guarantees reasonable returns and can reduce competitive pressure) and the goal of fostering innovation and competition. Despite these challenges, the public utility framework offers important conceptual advantages for thinking about essential infrastructure governance, and hybrid models that combine elements of utility oversight with market competition warrant serious exploration.

Federal investment in open-source, modular EHR components could further promote competition by enabling modular innovation, allowing hospitals to adopt interoperable modules for scheduling, billing, or clinical notes without committing to an all-or-nothing vendor ecosystem. Such modularity would reduce switching costs and foster a more pluralistic health IT market. However, creating viable open-source alternatives to Epic's comprehensive platform requires confronting substantial resource and coordination challenges. Epic has invested billions of dollars and decades of development effort into its

current platform [35,36]. Creating functionally equivalent open-source components would require sustained federal investment likely measured in hundreds of millions to billions of dollars, plus ongoing maintenance and security updates. The Veterans Health Administration's VistA system, often cited as a successful open-source EHR, required massive federal investment and still struggled with modernization, ultimately leading to a controversial decision to replace it with Cerner (now Oracle Health) [77]. Coordination challenges are equally daunting: achieving consensus among diverse stakeholders (clinicians, hospitals, payers, regulators) on requirements and priorities, maintaining consistent quality across modular components developed by different teams, and ensuring seamless interoperability between modules from multiple sources. Open-source healthcare IT projects have historically struggled with sustainability, as maintaining complex clinical software requires not just initial development but continuous adaptation to regulatory changes, security threats, and evolving clinical practice. Commercial support ecosystems must develop around open-source platforms for most hospitals to adopt them, which requires market conditions that do not currently exist. Despite these significant obstacles, open-source approaches deserve continued investment and experimentation, particularly for well-defined modular components where the coordination challenges are more manageable. Success will require realistic timelines, substantial resources, and recognition that open-source alone cannot solve the structural market problems—it must be paired with interoperability mandates, reduced switching costs, and other reforms that allow modular alternatives to compete fairly.

Long-term reform must also address the usability crisis. EHR platforms should be redesigned collaboratively with clinicians, patients, and human factors experts, shifting away from billing optimization toward workflows that enhance safety, efficiency, and patient engagement. Public funding or targeted tax credits could incentivize hospitals to pilot these next-generation systems, creating viable alternatives to the entrenched Epic model. Finally, reform requires leadership renewal and interdisciplinary governance. Solving systemic issues in digital health cannot be left to the same institutions and silos that created them. A new generation of leaders (including nurses, allied health providers, human factors engineers, social scientists, and patients) must be empowered to reimagine how health IT is governed and for whom it is designed.

## Stakes

Epic's dominance has created a system in which decisions about access, interoperability, and innovation are effectively privatized, with minimal oversight and few competitive alternatives. The stakes, therefore, extend far beyond software procurement. When a single company controls the digital backbone of U.S. healthcare, the flow of patient information becomes governed by private incentives rather than public needs. In practice, Epic functions as a quasi-regulatory authority, setting the terms by which data are stored, exchanged, and monetized. This dynamic erodes democratic accountability, as foundational decisions about health information infrastructure occur outside the reach of public institutions. Concentrating such authority in a single, privately held corporation poses profound risks to transparency, equity, and long-term system resilience.

Market concentration has historically dampened innovation, and the EHR sector is no exception. Epic's dominance diminishes incentives to improve usability, interoperability, and clinician-centered design. Smaller vendors with promising technologies face nearly insurmountable barriers to entry, from prohibitively high switching costs to structural interoperability disadvantages. Without reform, the U.S. risks locking itself into a stagnant health IT ecosystem, where optimizing revenue cycle management takes precedence over improving patient outcomes or harnessing next-generation technologies such as AI and precision medicine.

At its core, the stakes are ultimately clinical. Information blocking, contractual coercion, and incomplete interoperability do not merely distort markets, they directly affect patients. Federal litigation filings allege that oncology patients lost access to records and hospitals experienced delays in data sharing that could alter treatment decisions, according to plaintiff complaints in ongoing cases [4]. Beyond individual harms, centralized control over health data creates systemic vulnerabilities: bottlenecks in innovation, inequities in access, and exposure to cyberattacks that could imperil national

health security. The trajectory of the U.S. EHR market suggests that without decisive intervention, Epic's dominance will become permanently entrenched. History shows that once monopolies consolidate around critical infrastructure—whether in telecommunications, energy, or transportation—reform becomes exponentially harder. Policymakers now face a narrowing window in which meaningful action remains possible.

The next frontier of this privatization is artificial intelligence. As Epic expands its control from health records to AI-enabled decision support and predictive analytics, the same structural forces that monopolized digital infrastructure now threaten to concentrate the algorithms that interpret it. When one private vendor owns both the data pipelines and the models trained on them, governance of healthcare's digital future shifts entirely outside public accountability. Without intervention, the United States risks not only privatizing its digital backbone but the very cognitive infrastructure—the decision-making intelligence—of its healthcare system.

## Conclusion

Epic Systems has built one of the most entrenched positions of market power in the history of healthcare information technology. Controlling over half of U.S. acute care hospital beds and dominating academic medical centers, Epic now serves as the de facto gatekeeper of health data for hundreds of millions of Americans. Its power rests not on technological superiority but on structural mechanisms that make defection prohibitively costly. Federal litigation further documents active exclusionary practices, from information blocking to product tying, reinforcing that Epic's dominance is sustained by deliberate tactics as well as market structure. Leaving the digital backbone of healthcare in the hands of a single private vendor risks stagnation in innovation, inequities in access, and vulnerabilities in governance. Patient care is already compromised by interoperability barriers and information blocking. Without intervention, Epic's monopoly is likely to harden into permanence, embedding a private chokepoint at the center of U.S. healthcare infrastructure. The U.S. now faces a choice: allow one vendor to dictate the terms of healthcare's digital future, or reclaim health information as a public resource, governed in the interests of patients, clinicians, and society. Confronting Epic's monopoly is not merely a matter of market fairness, it is essential for the future of healthcare innovation, governance, and patient care.

## Author contributions

**Conceptualization:** Rawan Abulibdeh, Matthew G. Crowson, Molly J Douglas, Mena Ramos, Leo Anthony Celi.

**Investigation:** Rawan Abulibdeh.

**Methodology:** Rawan Abulibdeh.

**Project administration:** Leo Anthony Celi.

**Supervision:** Leo Anthony Celi.

**Visualization:** Matthew G. Crowson.

**Writing – original draft:** Rawan Abulibdeh.

**Writing – review & editing:** Rawan Abulibdeh, Matthew G. Crowson, Molly J Douglas, Mena Ramos, Noelle N. Saillant, Leo Anthony Celi.

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
