## [Decision Letter · Decision Letter 0]

25 Dec 2025

Response to Reviewers
Revised Manuscript with Track Changes
Manuscript
**Journal Requirements:**

1. Please send a completed 'Competing Interests' statement, including any COIs declared by your co-authors. If you have no competing interests to declare, please state "The authors have declared that no competing interests exist". Otherwise please declare all competing interests beginning with the statement "I have read the journal's policy and the authors of this manuscript have the following competing interests:"

2. We ask that a manuscript source file is provided at Revision. Please upload your manuscript file as a .doc, .docx, .rtf or .tex.

3. Please provide an Author Summary. This should appear in your manuscript between the Abstract (if applicable) and the Introduction, and should be 150–200 words long. The aim should be to make your findings accessible to a wide audience that includes both scientists and non-scientists. Sample summaries can be found on our website under Submission Guidelines:

https://journals.plos.org/digitalhealth/s/submission-guidelines#loc-parts-of-a-submission

4. We have noticed that you have cited Supporting Information in your manuscript. However, there are no corresponding files uploaded to the submission. Please upload them as separate files with the item type 'Supporting Information'.

5. We have noticed that you have uploaded Supporting Information files, but you have not included a list of legends. Please add a full list of legends for your Supporting Information files after the references list.

6. We note that your Data Availability Statement is currently as follows: “All data are in the manuscript and/or supporting information files.”

**Additional Editor Comments (if provided):**
**Reviewers' Comments:**

**Comments to the Author**

1. Does this manuscript meet PLOS Digital Health’s publication criteria?

Reviewer #1: Yes

Reviewer #2: Yes

Reviewer #3: Yes

2. Has the statistical analysis been performed appropriately and rigorously?

Reviewer #1: N/A

Reviewer #2: Yes

Reviewer #3: Yes

3. Have the authors made all data underlying the findings in their manuscript fully available (please refer to the Data Availability Statement at the start of the manuscript PDF file)?

Reviewer #1: Yes

Reviewer #2: Yes

Reviewer #3: Yes

4. Is the manuscript presented in an intelligible fashion and written in standard English?

Reviewer #1: Yes

Reviewer #2: Yes

Reviewer #3: Yes

Reviewer #1: This manuscript provides a timely and comprehensive analysis of structural power in the U.S. EHR market. The manuscript is well written, conceptually rich, and raises important policy-relevant questions. The Introduction clearly and effectively articulates the overarching problems addressed in this study, and I found the stepwise transformation framework particularly engaging. I believe this work will be of interest to a broad audience in digital health.

I recommend acceptance of this manuscript, subject to minor editorial revisions. As part of the publication process, I encourage the authors to review the manuscript carefully and address the following editable points:

1. Introduction, line 15: There is a typographical error — “HITEC” should be corrected to “HITECH.”

2. Use of acronyms: Acronyms such as FHIR appear before their full terms are defined (e.g., FHIR first appears at line 109, while the full term is introduced later at line 157). Similarly, TEFCA appears at line 109, while the full term is defined at line 378. The full term should be introduced at its first mention.

Overall, this is a strong and engaging manuscript. Addressing these minor editorial issues will further improve clarity and coherence, and strengthen the manuscript’s contribution to ongoing discussions on EHR market structure and policy reform.

Reviewer #2: This manuscript presents a timely and well-executed analysis of electronic health record (EHR) market concentration in the United States, with Epic Systems examined as a central case. The topic is highly relevant to the scope of PLOS Digital Health and addresses important questions related to interoperability, governance, and public accountability in digital health infrastructure.

The analytical approach is appropriate and rigorous. The authors effectively integrate quantitative market concentration metrics with qualitative synthesis of peer-reviewed literature, regulatory history, legal filings, and international case examples. The statistical analyses are sound, clearly described, and properly interpreted, and the conclusions are well supported by the presented evidence without overstatement.

Data sources are transparently cited and publicly available, and the Data Availability Statement accurately reflects the nature of the underlying data. Ethical and publication considerations, including ongoing litigation and the use of Epic-derived clinical data, are handled carefully and responsibly.

The manuscript is clearly written, well structured, and accessible to an interdisciplinary audience. Overall, this study makes a valuable contribution to the digital health and health policy literature, and I recommend acceptance without revision.

Reviewer #3: Reviewer Comments to the Author

General Assessment: This paper presents an exceptionally timely and rigorous analysis of market concentration in the U.S. EHR sector. The authors effectively argue that Epic’s dominance is driven not merely by product superiority but by structural market forces, utilizing international case studies (Norway, Denmark, UK) to highlight how U.S.-specific market structures have facilitated this consolidation. The proposed policy reforms are well-reasoned and highly relevant to current legislative discussions.

To further strengthen the manuscript and ensure precision, a minor revision is required.

Specific Comments:

1. Clarification of Legal Terminology regarding Pending Litigation

Context: In the "Stakes" section (Line 873), the manuscript notes that "Federal litigation has documented oncology patients losing access...".

Comment: While the author likely refers to plaintiff complaints, the phrase "litigation has documented" risks misrepresenting unproven allegations as adjudicated facts. Since Particle Health v. Epic and CureIS v. Epic are ongoing, I recommend qualifying these statements with phrases such as "allegedly," "claims that," or "according to the filings" to strictly maintain the manuscript's legal precision and neutrality.

**Do you want your identity to be public for this peer review?** For information about this choice, including consent withdrawal, please see our Privacy Policy

Reviewer #1: No

Reviewer #2: No

Reviewer #3: No

**Figure resubmission:**

**Reproducibility:** To enhance the reproducibility of your results, we recommend that authors of applicable studies deposit laboratory protocols in protocols.io, where a protocol can be assigned its own identifier (DOI) such that it can be cited independently in the future. Additionally, PLOS ONE offers an option to publish peer-reviewed clinical study protocols. Read more information on sharing protocols at https://plos.org/protocols?utm_medium=editorial-email&utm_source=authorletters&utm_campaign=protocols

---

## [Decision Letter · Decision Letter 1]

26 Feb 2026

A problem of Epic proportion

PDIG-D-25-01138R1

Dear Dr. Celi,

We are pleased to inform you that your manuscript 'A problem of Epic proportion' has been provisionally accepted for publication in PLOS Digital Health.

Best regards,

Dukyong Yoon

Section Editor

PLOS Digital Health

**Additional Editor Comments (if provided):**

**Reviewer Comments (if any, and for reference):**

Reviewer's Responses to Questions

**Comments to the Author**

Reviewer #3: All comments have been addressed

publication criteria?

Reviewer #3: Yes

3. Has the statistical analysis been performed appropriately and rigorously?

Reviewer #3: Yes

4. Have the authors made all data underlying the findings in their manuscript fully available (please refer to the Data Availability Statement at the start of the manuscript PDF file)?

Reviewer #3: Yes

5. Is the manuscript presented in an intelligible fashion and written in standard English?

Reviewer #3: Yes

Reviewer #3: I have reviewed the revised manuscript and the authors' response to the comments. The authors have satisfactorily addressed all previous concerns. I have no further comments and believe the paper is now ready for publication.

**Do you want your identity to be public for this peer review?** For information about this choice, including consent withdrawal, please see our Privacy Policy

Reviewer #3: No
